# Mechanical on-chip microwave circulator

S. Barzanjeh[1], M. Wulf[1], M. Peruzzo[1], M. Kalaee[2,3], P.B. Dieterle[2,3], O. Painter[2,3] & J.M. Fink [1]

Nonreciprocal circuit elements form an integral part of modern measurement and communication systems. Mathematically they require breaking of time-reversal symmetry, typically achieved using magnetic materials and more recently using the quantum Hall effect, parametric permittivity modulation or Josephson nonlinearities. Here we demonstrate an on-chip magnetic-free circulator based on reservoir-engineered electromechanic interactions. Directional circulation is achieved with controlled phase-sensitive interference of six distinct electro-mechanical signal conversion paths. The presented circulator is compact, its silicon-on-insulator platform is compatible with both superconducting qubits and silicon photonics, and its noise performance is close to the quantum limit. With a high dynamic range, a tunable bandwidth of up to 30 MHz and an in situ reconfigurability as beam splitter or wavelength converter, it could pave the way for superconducting qubit processors with multiplexed on-chip signal processing and readout.

[1] Institute of Science and Technology Austria, 3400 Klosterneuburg, Austria. [2] Kavli Nanoscience Institute and Thomas J. Watson, Sr., Laboratory of Applied Physics, California Institute of Technology, Pasadena, CA 91125, USA. [3] Institute for Quantum Information and Matter, California Institute of Technology, Pasadena, CA 91125, USA. Correspondence and requests for materials should be addressed to S.B. (email: shabir.barzanjeh@ist.ac.at) or to J.M.F. (email: jfink@ist.ac.at)

Nonreciprocal devices are quintessential tools to suppress spurious modes, interferences and unwanted signal paths[1]. More generally, circulators can be used to realize chiral networks[2] in systems where directional matter-light coupling is not easily accessible. In circuit quantum electro-dynamics[3] circulators are used for single-port coupling or as isolators to protect the vulnerable cavity and qubit states from electromagnetic noise and strong parametric amplifier drive tones. State-of-the-art passive microwave circulators are based on magneto-optic effects that require sizable magnetic fields[4,5], incompatible with ultra-low loss superconducting circuits. Due to the design principle their size is at least on the order of the wavelength and during manufacturing they need to be tuned and optimized one by one. Commercial circulators can therefore not be integrated on-chip causing additional losses and forming a major roadblock towards a fully integrated quantum processor based on superconducting qubits.

Many recent theoretical and experimental efforts have been devoted to overcome these limitations both in the optical[6-8] and microwave regimes[9-18]. In parallel, the rapidly growing field of optomechanical and electromechanical systems has shown promising potential for applications in quantum information

processing and communication, in particular for microwave to optical conversion[19,20] and amplification[21]. Very recently, several theoretical proposals[22-24] have pointed out that reservoir-engineered optomechanical systems[25] can lead to nonreciprocity and first isolators have just been demonstrated in the optical domain[26-28].

Here we present an on-chip microwave circulator using a frequency tuneable silicon-on-insulator electromechanical system[29] that is compatible with superconducting qubits[30]. The device can be reconfigured in situ as a filter, splitter, mixer, isolator or circulator. For the first mechanically mediated microwave circulator, we achieve an isolation of up to 24–38 dB, compared to total losses of 4.7–8.5 dB and at most 4–7 added noise quanta over an instantaneous bandwidth of 630 Hz.

## Results

**Device characterization and tunability.** The main elements of the microchip circulator device are shown in Fig. 1a, b. The circuit comprises three high-impedance spiral inductors ($L_i$) capacitively coupled to the in-plane vibrational modes of a dielectric nanostring mechanical resonator. The nanostring

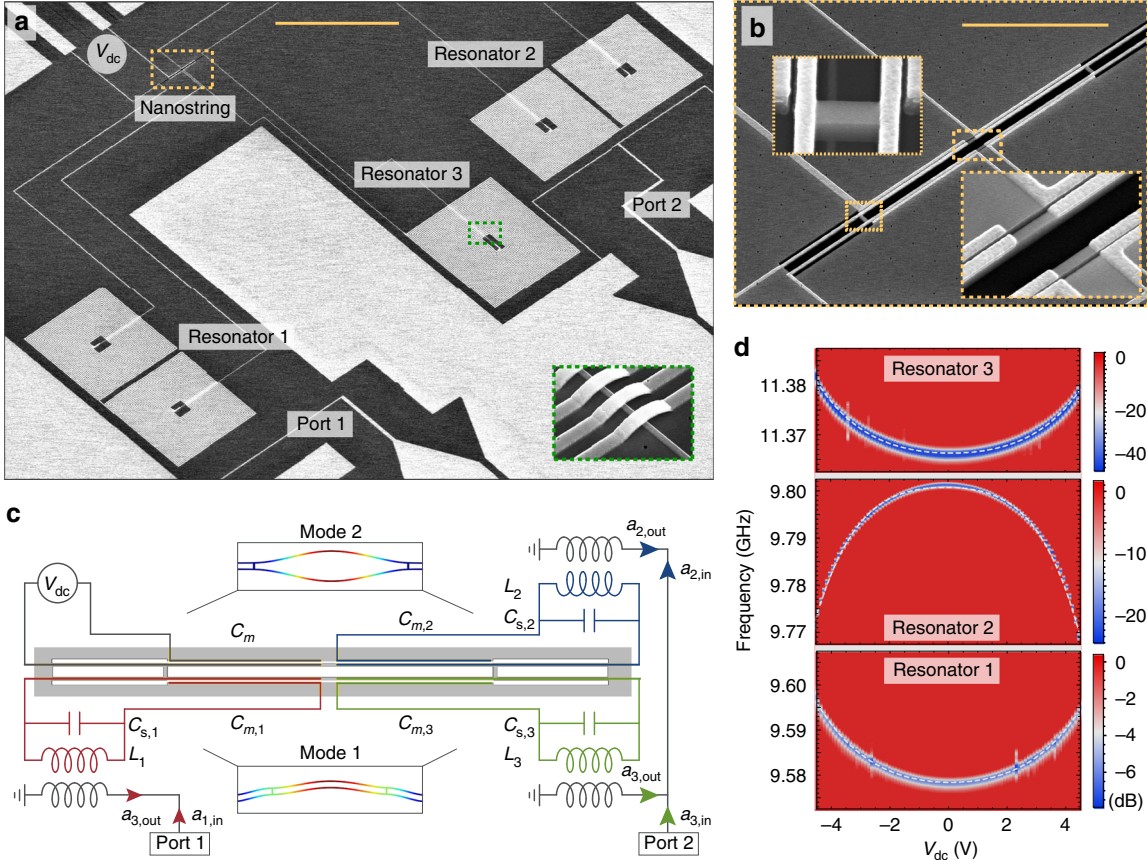

**Fig. 1** Microchip circulator and tunability. **a** Scanning electron micrograph of the electromechanical device including three microwave resonators, two physical ports labelled 1 and 2, one voltage bias input ($V_{dc}$) on the top left. The inset shows the spiral inductor cross-overs in the green dashed boxed area. The total device area is 0.3 mm by 0.45 mm. Scale bar (yellow) corresponds to 100 μm. **b** Enlarged view of the silicon nanostring mechanical oscillator with four vacuum-gap capacitors coupled to the three coil inductors and one voltage bias input. Insets show details of the nanobeam as indicated by the dashed and dotted rectangles. Scale bar (yellow) corresponds to 10 μm. **c** Electrode design and electrical circuit diagram of the device. The input modes $a_{i,in}$ couple inductively to the microwave resonators with inductances $L_i$, coil capacitances $C_i$, additional stray capacitances $C_{s,i}$ and the motional capacitances $C_{m,i}$. The reflected tones $a_{i,out}$ pass through a separate chain of amplifiers each, and are measured at room temperature using a phase-locked spectrum analyzer (not shown). The simulated displacement of the lowest frequency in-plane flexural modes of the nanostring are shown in the two insets. Colour indicates relative displacement. **d** Resonator reflection measurement of the three microwave resonators of an identical device, as a function of the applied bias voltage and a fit (dashed lines) to $\Delta\omega = \alpha_1 V^2 + \alpha_2 V^4$ with the tunabilties $\alpha_1/2\pi = 0.53$ MHz/V$^2$ and $\alpha_2/2\pi = 0.05$ MHz/V$^4$ with a total tunable bandwidth of 30 MHz for resonator 2 at 9.8 GHz

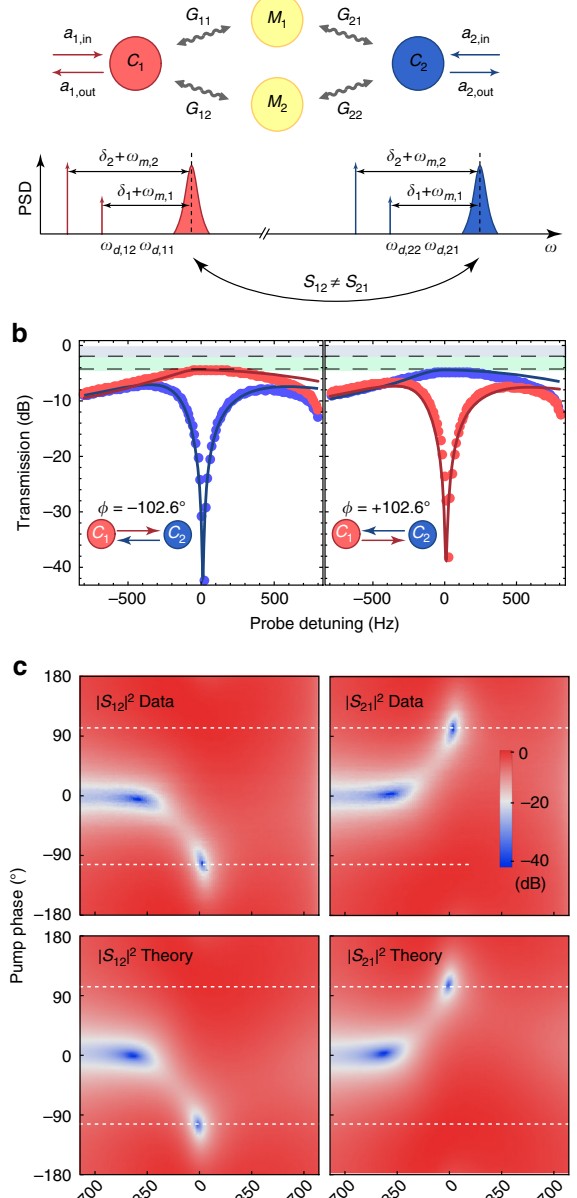

**Fig. 2** Electromechanical isolator. **a** Mode coupling diagram for electromechanically induced nonreciprocity. Two microwave cavities ($C_1$ and $C_2$) are coupled to two mechanical modes ($M_1$ and $M_2$) with the electromechanical coupling rates $G_{ij}$ (where $i, j = 1, 2$), inducing two distinct signal conversion paths. Power spectral density (PSD) of the two microwave cavities and arrows indicating the frequency of the four microwave pump tones slightly detuned by $\delta_i$ from the lower motional sidebands of the resonances. All four pumps are phase-locked while the signal tone is applied. Only one of the microwave source phases is varied to find the optimal interference condition for directional transmission between ports 1 and 2. **b** Measured power transmission (dots) in forward $|S_{21}|^2$ (cavity 1 → cavity 2) and backward directions $|S_{12}|^2$ (cavity 2 → cavity 1) as a function of probe detuning for two different phases $\phi = \pm 102.6°$. The solid lines show the results of the coupled-mode theory model discussed in the text. Grey shaded areas denote cavity loss and green shaded areas insertion loss. **c** Experimental data and theoretical model of measured transmission coefficients $|S_{12}|^2$ and $|S_{21}|^2$ as a function of signal detuning and pump phase $\phi$. Dashed lines indicate the line plots for the two phases $\phi = \pm 102.6°$ as shown in **b**

oscillator consists of two thin silicon beams that are connected by two symmetric tethers and fabricated from a high resistivity silicon-on-insulator device layer[29]. Four aluminium electrodes are aligned and evaporated on top of the two nanostrings, forming one half of the vacuum gap capacitors that are coupled to three microwave resonators and one DC voltage bias line as shown schematically in Fig. 1c (see Supplementary Tables 1 and 2 for details). The device is mounted on the mixing chamber plate of a cryogen-free dilution refrigerator at a temperature of $T_f = 10$ mK and all incoming lines are strongly filtered and attenuated to suppress Johnson and phase noises.

The voltage bias line can be used to generate an attractive force that pulls the nanobeam and tunes the operating point frequencies of the device[31]. Figure 1d shows the measured resonance frequency change as a function of the applied bias voltage $V_{dc}$. As expected, resonators 1 and 3 are tuned to higher frequency due to an increased vacuum gap, whereas resonator 2 is tuned to lower frequency. At a large tunable bandwidth of up to 30 MHz, as obtained for resonator 2, the ability to excite the motion directly and to modulate the electromechanical coupling in situ represents an important step towards new optomechanical experiments and more practicable on-chip reciprocal and nonreciprocal devices.

As a first step, we carefully calibrate and characterize the individual electromechanical couplings and noise properties, similar to ref. [32]. The measured thermalization temperature of the two mechanical modes are 18 and 25 mK and the final occupancy for the reported device reached as low as 0.6 and 2 quanta for standard motional sideband cooling. We then measure the bidirectional frequency conversion between two microwave resonator modes as mediated by one mechanical mode[33]. The incoming signal photons can also be distributed to two ports with varying probability as a function of the parametric drive strength and in direct analogy to a tunable beam splitter. We present the relevant sample parameters in Supplementary Tables 1 and 2, the theoretical analysis in Supplementary Note 2 and experimental results of this bidirectional frequency conversion process in Supplementary Fig. 1.

**Theoretical model.** Directionality is achieved by engaging the second mechanical mode, a method that was developed in parallel to this work[34,35] for demonstrating nonreciprocity in single-port electromechanical systems. In fact, creating a parametric coupling between the two electromagnetic and the two mechanical modes by four microwave pumps with frequencies slightly detuned from the lower motional sidebands of the resonances, creates two paths for exchanging photons, as shown in Fig. 2a. Nonreciprocity appears when these paths interfere destructively, leading to breaking the symmetry between the two directions. For a detailed description, we begin with the theoretical model describing two microwave cavities with resonance frequencies $\omega_i$ and total linewidths $\kappa_i$ with $i = 1, 2$ parametrically coupled to two distinct modes of a mechanical resonator with resonance frequencies $\omega_{m,j}$ and damping rates $\gamma_{m,j}$ with $j = 1, 2$. To establish the parametric coupling, we apply four microwave tones, with frequencies detuned by $\delta j$ from the lower motional sidebands of the resonances, as shown in Fig. 2a. In a reference frame rotating at the frequencies $\omega_i$ and $\omega_{m,j} + \delta j$, the linearized Hamiltonian in the resolved sideband regime ($\omega_{m,j} \gg \kappa_1, \kappa_2$) is given by ($\hbar = 1$)

$$H = -\sum_{j=1,2} \delta_j b_j^\dagger b_j + \sum_{i,j=1,2} G_{ij}\left(e^{i\phi_{ij}} a_i b_j^\dagger + e^{-i\phi_{ij}} a_i^\dagger b_j\right) \quad (1)$$

$$+ H_{off},$$

where $a_i(b_j)$ is the annihilation operator for the cavity $i$ (mechanics $j$), $G_{ij} = g_{0,ij}\sqrt{n_{ij}}$ and $g_{0ij}$ are the effective and vacuum electromechanical coupling rates between the mechanical mode $j$ and cavity $i$, respectively, while $n_{ij}$ is the total number of photons inside the cavity $i$ due to the drive with detuning $\Delta_{ij}$, and $\phi_{ij}$ is the relative phase set by drives. Here, $\Delta_{11} = \Delta_{21} = \omega_{m,1} + \delta_1$ and $\Delta_{22} = \Delta_{12} = \omega_{m,2} + \delta_2$ are the detunings of the drive tones with respect to the cavities and $H_{\mathrm{off}}$ describes the time-dependent coupling of the mechanical modes to the cavity fields due to the off-resonant drive tones. These additional coupling terms create cross-damping[36] and renormalize the mechanical modes, and can only be neglected in the weak coupling regime for $G_{ij}, \kappa_j \ll \omega_{m,j}, |\omega_{m,2} - \omega_{m,1}|$.

To see how the nonreciprocity arises, we use the quantum Langevin equations of motion along with the input–output theorem to express the scattering matrix $S_{ij}$ of the system described by the Hamiltonian (1), and relating the input photons $a_{\mathrm{in},i}(\omega_i)$ at port $i$ to the output photons $a_{\mathrm{out},j}(\omega_j)$ at port $j$ via $a_{\mathrm{out},i} = \sum_{j=1,2} S_{ij} a_{\mathrm{in},i}$ with $i = 1, 2$. The dynamics of the four-mode system described by Hamiltonian (1) is fully captured by a set of linear equations of motion as verified in Supplementary Notes 3–6. Solving these equations in the frequency domain, using the input–output relations, and setting $\phi_{22} = \phi$, $\phi_{11} = \phi_{21} = \phi_{12} = 0$, the ratio of backward to forward transmission reads

$$\lambda := \frac{S_{12}(\omega)}{S_{21}(\omega)} = \frac{\sqrt{C_{11}C_{21}}\,\Sigma_{m,2}(\omega) + \sqrt{C_{12}C_{22}}\,\Sigma_{m,1}(\omega)e^{i\phi}}{\sqrt{C_{11}C_{21}}\,\Sigma_{m,2}(\omega) + \sqrt{C_{12}C_{22}}\,\Sigma_{m,1}(\omega)e^{-i\phi}}. \quad (2)$$

Here, $\Sigma_{m,j} = 1 + 2i[(-1)^j\delta - \omega]/\gamma_{m,j}$ is the inverse of the mechanical susceptibility divided by the mechanical linewidth $\gamma_{m,j}$ and $C_{ij} = 4G_{ij}^2/(\kappa_i\gamma_{m,j})$ is the electromechanical cooperativity. Note that, in Eq. (2) we assume the device satisfies the impedance-matching condition on resonance, i.e., $S_{ii}(\omega = 0) = 0$, which can be achieved in the high-cooperativity limit ($C_{ij} \gg 1$).

Inspection of Eq. (2) reveals the crucial role of the relative phase between the drive tones $\phi$ and the detuning $\delta$ to obtain nonreciprocal transmission. When the cooperativities for all four electromechanical couplings are equal ($C_{ij} = \mathcal{C}$) then perfect isolation, i.e. $\lambda = 0$, occurs for

$$\tan[\phi(\omega)] = \frac{\delta(\gamma_{m,1} + \gamma_{m,2}) + \omega(\gamma_{m,2} - \gamma_{m,1})}{\gamma_{m,1}\gamma_{m,2}/2 - 2(\delta^2 - \omega^2)}. \quad (3)$$

Equation (3) shows that on resonance ($\omega = 0$) $\tan[\phi] \propto \delta$, highlighting the importance of the detuning $\delta$ to obtain nonreciprocity. Tuning all four drives to the exact red sideband frequencies ($\delta = 0$) results in bidirectional behaviour ($\lambda = 1$). At the optimum phase $\phi$ given by Eq. (3), $\omega = 0$, and for two mechanical modes with identical decay rates ($\gamma_{m,1} = \gamma_{m,2} = \gamma$) the transmission in forward direction is given by

$$S_{21} = -\sqrt{\eta_1\eta_2}\left[\frac{4i\delta(1 - 2i\delta/\gamma)}{\mathcal{C}\gamma\left(1 + \frac{1+4\delta^2/\gamma^2}{2\mathcal{C}}\right)^2}\right] \quad (4)$$

where $\eta_{1(2)} = \kappa_{\mathrm{ext},1(2)}/\kappa_{1(2)}$ is the resonator coupling ratio and $\kappa_i = \kappa_{\mathrm{int},i} + \kappa_{\mathrm{ext},i}$ is the total damping rate. Here $\kappa_{\mathrm{int},i}$ denotes the internal loss rate and $\kappa_{\mathrm{ext},i}$ the loss rate due to the cavity to waveguide coupling. Equation (4) shows that the maximum of the transmission in forward direction, $|S_{21}|^2 = \eta_1\eta_2[1 - (2\mathcal{C})^{-1}]$, occurs when $2\mathcal{C} = 1 + 4\delta^2/\gamma^2$ and for large cooperativities $\mathcal{C} \gg 1$. These conditions, as implemented in our experiment, enable the observation of asymmetric frequency conversion with strong isolation in the backward direction and small insertion loss in forward direction.

**Bidirectional wavelength conversion.** For bidirectional wavelength conversion, higher cooperativity enhances the bandwidth, as shown in Supplementary Note 2. In contrast, the bandwidth of the nonreciprocal conversion is independent of cooperativity and set only by the intrinsic mechanical linewidths $\gamma_{m,i}$, which can be seen in Eq. (2). This highlights the fact that the isolation appears when the entire signal energy is dissipated in the mechanical environment, a lossy bath that can be engineered effectively[25]. In the present case it is the off-resonant coupling between the resonators and the mechanical oscillator that modifies this bath. The applied drives create an effective interaction between the mechanical modes, where one mode acts as a reservoir for the other and vice versa. This changes both the damping rates and the eigenfrequencies of the mechanical modes.

It, therefore, increases the instantaneous bandwidth of the conversion and automatically introduces the needed detuning, which is fully taken into account in the theory.

**Two-port microwave isolator.** Using the on-chip electromechanical microwave circuit shown in Fig. 1a, we experimentally realize directional wavelength conversion between two superconducting coil resonators at $(\omega_1, \omega_2)/2\pi = (9.55, 9.82)$ GHz coupled to two different physical waveguide ports and measurement lines with $(\eta_1, \eta_2) = (0.74, 0.86)$. Here we use the two lowest-frequency vibrational in-plane modes of the mechanical resonator at $(\omega_{m,1}, \omega_{m,2})/2\pi = (4.34, 5.64)$ MHz with intrinsic damping rates $(\gamma_{m,1}, \gamma_{m,2})/2\pi = (4, 8)$ Hz. The vacuum electromechanical coupling strengths for these mode combinations are $(g_{0,11}, g_{0,12}, g_{0,21}, g_{0,22})/2\pi = (33, 34, 13, 31)$ Hz. The microwave resonators are driven with four coherent microwave sources with powers $(P_{11}, P_{12}, P_{21}, P_{22}) = (-73.3, -68.7, -66.9, -67.4)$ dBm at the device inputs that correspond to the single cavity–single mechanical cooperativities $(C_{11}, C_{12}, C_{21}, C_{22}) = (47, 43.8, 41.9, 56.9)$. Transmission parameters are measured by using a weak probe signal with a signal power of only $-117$ dBm at the device inputs.

Figure 2b shows the measured transmission of the wavelength conversion in the forward $|S_{21}|^2$ and backward directions $|S_{12}|^2$ as a function of probe detuning for two different phases as set by one out of the four phase-locked microwave drives. At $\phi = -102.6°$ and over a bandwidth of 518 Hz, we measure high transmission from cavity 1 to 2 with an insertion loss of 2.4 dB due to finite input matching and a resonator loss of 1.9 dB due to finite intrinsic resonator linewidths. In the backward direction, the transmission is suppressed by up to 40.4 dB. Likewise, at the positive phase of $\phi = 102.6°$ the transmission from cavity 1 to 2 is suppressed while the transmission from cavity 2 to 1 is high. In both cases, we observe excellent agreement with theory (solid lines). Figure 2c shows the $S$ parameters for the whole range of phases $\phi$, which are symmetric and bidirectional around $\phi = 0$. We find excellent agreement with theory over the full range of measured phases with <10% deviation to independently calibrated drive photon numbers and without any other free parameters.

**Extension to a microwave circulator.** The described two-port isolator can be extended to an effective three-port device by parametrically coupling the third microwave resonator capacitively to the dielectric nanostring, as shown in Fig. 1a. The third resonator at a resonance frequency of $\omega_3/2\pi = 11.30$ GHz is coupled to the waveguide with $\eta_3 = 0.52$ and to the two in-plane mechanical modes with $(g_{0,31}, g_{0,32})/2\pi = (22, 45)$ Hz. Similar to the isolator, we establish a parametric coupling between cavity and mechanical modes using six microwave pumps with

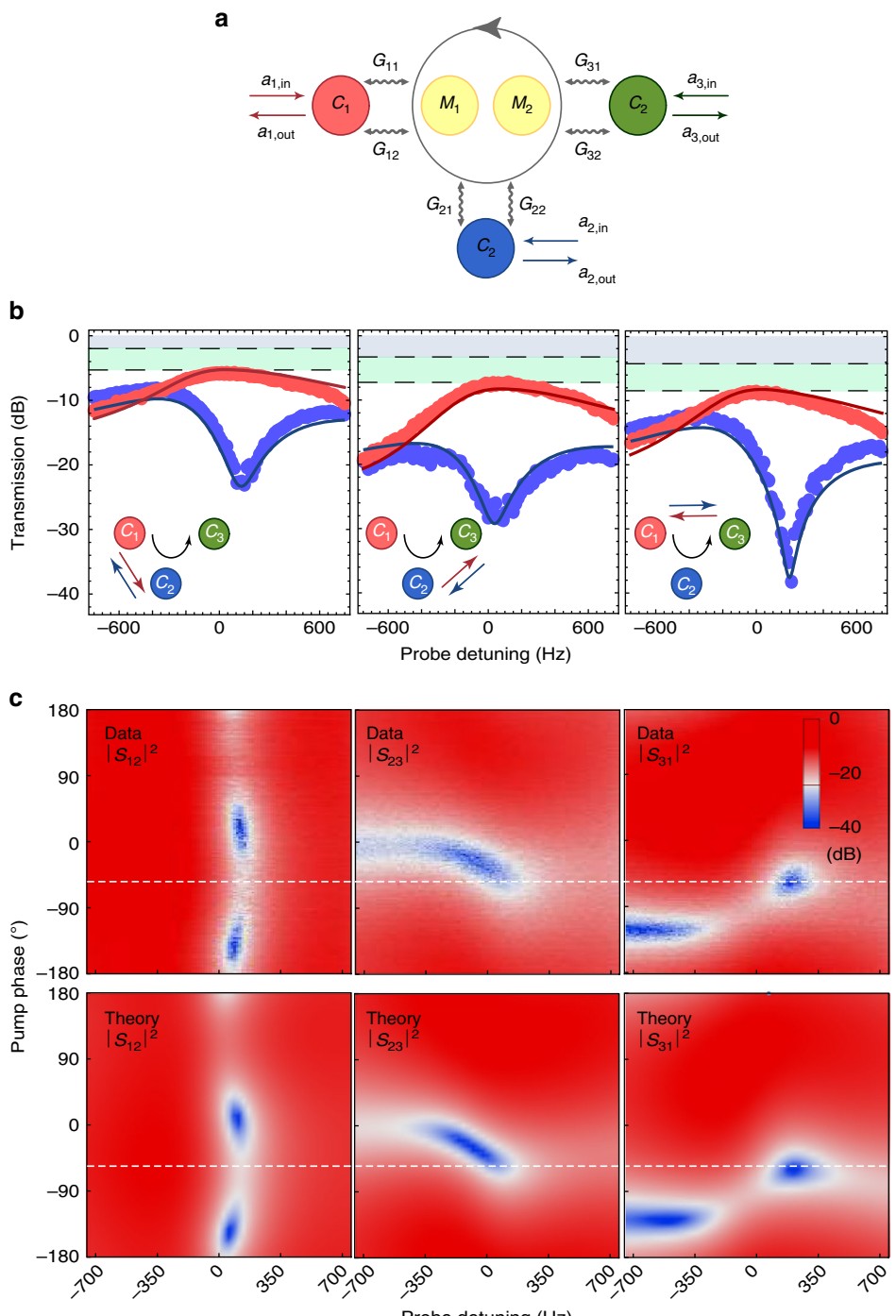

**Fig. 3** Electromechanical circulator. **a** Mode coupling diagram describing the coupling between three microwave cavities ($C_1$, $C_2$ and $C_3$) and two mechanical modes ($M_1$ and $M_2$) with electromechanical coupling rates $G_{ij}$ (where $i = 1, 2, 3$ and $j = 1, 2$), creating a circulatory frequency conversion between the three cavity modes, as indicated by the grey arrow. **b** Measured power transmission in forward $|S_{21}|^2$, $|S_{32}|^2$ and $|S_{13}|^2$ (red dots) and backward directions $|S_{12}|^2$, $|S_{23}|^2$ and $|S_{31}|^2$ (blue dots) as a function of probe detuning for a pump phase $\phi = -54°$. The solid lines show the prediction of the coupled-mode theory model discussed in the text. The inset shows the signal propagation between the three resonator modes and the black arrow indicates the circulator direction. Grey shaded areas denote cavity loss and green shaded areas insertion loss. **c** Measured $S$ parameters and theoretical model as a function of detuning and pump phase. Dashed lines indicate the line plot positions shown in **b**

frequencies slightly detuned from the lower motional sidebands of the resonances, which for certain pump phase combinations can operate as a three-port circulator for microwave photons, see Fig. 3a. The pump powers at the device inputs are ($P_{11}$, $P_{12}$, $P_{21}$, $P_{22}$, $P_{31}$, $P_{32}$) = (−72.5, −69, −67.5, −68, −69, −70) corresponding

to the single resonator cooperativities ($C_{11}$, $C_{12}$, $C_{21}$, $C_{22}$, $C_{31}$, $C_{32}$) = (56.5, 40.9, 35, 49.6, 99.2, 49, 6). Using an additional microwave source as a weak probe signal with a signal power of only −117 dBm at the device inputs, we measure the power transmission between all ports and directions as shown in Fig. 3b

for a single fixed phase of $\phi = -54^\circ$, optimized experimentally for forward circulation.

At this phase, we see high transmission in the forward direction $S_{21,32,13}$ with an insertion loss of (3.8, 3.8, 4.4) dB, due to imperfect input matching, and an isolation in the backward direction $S_{12,23,31}$ of up to (18.5, 23, 23) dB over a bandwidth of 628.5 Hz. The full dependence of the circulator scattering parameters on the drive phase is shown in Fig. 3c, where we see excellent agreement with theory. The added noise photon number of the device is found to be ($n_{add,21}$, $n_{add,32}$, $n_{add,13}$) = (4, 6.5, 3.6) in the forward direction and ($n_{add,12}$, $n_{add,23}$, $n_{add,31}$) = (4, 4, 5.5) in the backward direction, limited by the thermal occupation of the mechanical modes and discussed in more detail in Supplementary Fig. 3.

## Discussion

In conclusion, we demonstrated a frequency tunable and in situ reconfigurable signal processing device that can act as a filter, wavelength converter, beam splitter, isolator or circulator for microwave photons. The circulator is highly directional and operates with relatively low loss and added noise. Improvements of the circuit design and fabrication will help to increase the instantaneous bandwidth and decrease the insertion losses. Compared to the so far significantly higher bandwidth Josephson devices, a mechanical approach is insensitive to magnetic field noise and offsets, mechanical devices have higher dynamical range due to the smaller nonlinearity, and well-confined mechanical modes are typically less prone to parasitic coupling when integrated in larger systems. In addition, mechanical systems have the potential for hybrid microwave and optical signal processing, in particular for non-reciprocity between microwave and optical propagating fields. The presented external voltage bias offers new ways to achieve directional amplification and squeezing of microwave fields in the near future.

**Data availability**. The data that support the findings of this study are available from the corresponding author upon reasonable request.

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

## Acknowledgements

We thank Nikolaj Kuntner for the development of the Python virtual instrument panel and Georg Arnold for supplementary device simulations. This work was supported by IST Austria and the European Union's Horizon 2020 research and innovation program under grant agreement No 732894 (FET Proactive HOT). S.B. acknowledges support from the European Union's Horizon 2020 research and innovation program under the Marie Sklodowska Curie grant agreement No 707438 (MSC-IF SUPEREOM). OJP acknowledges support from the AFOSR-MURI Quantum Photonic Matter, the Institute for Quantum Information and Matter, an NSF Physics Frontiers Center (grant PHY-1125565) with support of the Gordon and Betty Moore Foundation, and the Kavli Nanoscience Institute at Caltech.

## Author contributions

S.B. and J.M.F. conceived the ideas for the experiment. S.B. developed the theoretical model, performed and analysed the measurements. S.B., M.W., M.P. and J.M.F. designed the microwave circuit and built the experimental setup. M.K., P.B.D., J.M.F. and O.P.

designed the mechanical nanobeam oscillator. J.M.F. and M.K. fabricated the sample. P.B.D. and O.P. contributed to sample fabrication. S.B. and J.M.F. wrote the manuscript. J.M.F. supervised the research.

## Additional information

**Competing interests:** The authors declare no competing financial interests.

