## [Peer Review File · Nature Communications]

Reviewers' comments:

Reviewer #1 (Remarks to the Author):

In the manuscript "Mechanical On-Chip Microwave Circulator", the authors have reported the experimental realization of frequency tunable microwave isolator/circulator. The experiment is indeed very interesting and the data well understood and modeled in detail. It would generate a lot of excitement, as an on-chip microwave isolator/circulator which could even work at the single "photon" level. In my opinion, the manuscript is suitable for publication in Nature Communication, after the authors have addressed the following comments and questions:

1. Can the authors discuss more details about the advantages of such devices, especially for the quantum information processing? The discussion in the manuscript is too simple. It will be really helpful if the authors can estimate some parameters for experiment, i.e. temperature, cooperativity.
2. What's the bandwidth of the isolator/circulator? Is there any more application of such small bandwidth especially for the microwave system?
3. What's the input power to drive the mechanics and get higher cooperativity? Is there any nonlinear effect?
4. The figures are small and illegible to read. It can be improved to one row. And what's the unit of the Fig.2b & 3b?

Reviewer #2 (Remarks to the Author):

In the manuscript "Mechanical On-Chip Microwave Circulator" the authors report on an experiment based on the existing theoretical ideas, to implement a microwave isolator/circulator. Integrating optomechanics with superconducting qubits, by using the same microwave technology, is interesting and could provide new possibilities. This work builds on the existing ideas and experiments in cavity optomechanics (see ref. 26 and refs within). The authors demonstrate that not only the operating frequency of the device can be tuned, but also the direction of the isolation/circulation can be controlled. The experimental results are in a good agreement with authors' theoretical analysis. I think if the authors address the issues below, publishing the paper in Nature communication can benefit a wide range of audience.

- 1) In the second paragraph, where limitations of the existing circulators based on magneto-optic effects have been discussed, no reference is presented.
- 2) I think it helps if the authors clarify what the optomechanical and electromechanical couplings refer to in their setup.
- 3) While the physics of non-reciprocity is based on reservoir engineering and the presence of mechanical loss, the intuitive picture behind non-reciprocal mechanism is postponed after the presentation of formalism and the results. I think it's helpful to present the physical picture first, and then delve into a mathematical description.
- 4) It seems that in insets of Fig 3b the black arrow indicates the circulation direction. It would be useful to mention it in the caption.

Reviewer #3 (Remarks to the Author):

The primary innovation in the paper "Mechanical On-Chip Microwave Circulator" is the first experimental electro-mechanical circulator that is potentially chip-compatible with other cryogenic microwave components. This is in contrast to the various demonstrations of opto- (refs 24-26) and electro-mechanical (refs 28 & 29, apparently 29 is now in print) _isolators_, which are two port non-reciprocal devices, as opposed to three.

Not without good reason, non-reciprocal components are having a bit of a "moment" right now in integrated micro- and quantum-devices in the optical and RF domains, due to the limitations of the standard technologies based on permanent magnets, as mentioned in the manuscript. Thus, this paper opens the door to a new technological approach (i.e. mechanical) to achieving non-reciprocity, and has the potential to be of interest to a fairly wide audience in integrated micro- and quantum-devices in the optical and RF domains. So I feel the topic matter and this paper in general deserves to be published in a broad journal. The main limitation of the paper is that the circulator's performance is extremely poor by almost every metric compared to already proven Joseph junction-based RF circulators (refs. 4 & 5 are two, but not the only examples), and it's not at all clear how the mechanical approach could ever out compete them. So, it deserves publication for its basic science, even if I am very pessimistic about its technological impact.

Assuming the authors can adequately address my questions and comments further below, I would generally support publication in Nature Communications.

1) A small thing, but I generally prefer to describe devices such as this one as "electro-mechanics" not "opto-mechanics." Here the authors sort of use the terms interchangeably, which is not uncommon in the field because they are so closely linked, but I wonder if it might prove confusing to other readers.

2) Fig. 1c needs to be redrawn to make the electrical connectivity much clearer. I cannot tell at all what is electrically connected to what in the nanowire device and the text didn't help me understand what I couldn't see in the figure. In particular, I cannot tell at all where the Vdc is applying its potential. I am very confused why "As expected, resonators 1 and 3 are tuned to higher frequency due to an increased vacuum gap while resonator 2 is tuned to lower frequency." As far as I can tell in Fig. 1c, resonators 2 & 3 look identical.

3) In figure 1a, it looks like there are two additional inductively coupled ports. What are these for?

4) Please state the overall footprint area of the device.

5) If the cooperativity employed in the main text figures was stated, I missed it. Similarly, Please also state the pump power required to achieve this cooperativity in dBm. This required power metric is helpful in comparing this technology to others, such as the JJ devices.

6) What is the relationship between the theory discussed at length in this paper to that described in the theory papers referenced in refs 6, 22, & 23? What is the relationship between the theory in this paper and the theory in the JJ-based components? For example, ref. 4 also involves parametric frequency conversion between the three port modes and two internal modes to achieve circulation. Is this approach and that one formally equivalent?

7) I want to see not just the forward and backward scattering matrix elements, but S11, S22, & S33 as well. These input reflections are as important as the isolation for a usable device.

8) Similarly, I want to see an output PSD of all three ports while the component is circulating.

Again, as a practical matter, S-parameters are not the only thing you care about in a circulator, _especially_ an active one. This device employs six pump tones that are orders of magnitude stronger than the signal tones and merely MHzs from the signal carriers. It is critically important to know how bad all this RF leakage is.

9) I think it would benefit readers if the authors provided more of an honest assessment of this technology vis a vis other non-magnetic approaches. In particular, the JJ devices seems to outperform this one in just about every way, and it's not at all clear if this could ever change. For example, this device provided ~ 10 dB relative isolation over merely ~ 300 Hz, while ref. 4 (the first ever demonstration) had the same over 11 MHz. This component may be tuned by 30 MHz, while that one could be tuned up to 400 MHz. This one had ~ 4 quanta of added noise, while that one had 0.5 (quantum limited). This one requires six RF pumps, that one 3. Presumably the pump power required here is again orders of magnitude greater than in ref. 4, because the nonlinearity is so much weaker. If I had to guess, this mechanical technology might only be able to outperform the JJ ones in 1 dB compression point, again because the nonlinearity is so much weaker.

As for the proposed application of superconducting qubit readout, the bandwidth of those signals are measured in MHz, and carriers can easily vary by 100 MHz without a very well controlled process. Even with its better parameters, the limitations of ref. 4 have yielded little actual technological impact so far for this application either. I want to reiterate that publishable work just has to be innovative, rather than technologically competitive, but the authors can only help the field by comparing it to other technologies and offering more guidance on where improvements will be best made.

Reviewer #1 (Remarks to the Author):

In the manuscript "Mechanical On-Chip Microwave Circulator", the authors have reported the experimental realization of frequency tunable microwave isolator/circulator. The experiment is indeed very interesting and the data well understood and modeled in detail. It would generate a lot of excitement, as an on-chip microwave isolator/circulator which could even work at the single "photon" level. In my opinion, the manuscript is suitable for publication in Nature Communication, after the authors have addressed the following comments and questions:

We thank the referee for this positive comment about publication of our manuscript in Nature communication.

1a. Can the authors discuss more details about the advantages of such devices, especially for the quantum information processing? The discussion in the manuscript is too simple.

We have added a more detailed description of the purpose of a microwave circulator for quantum information processing in the first paragraph of the introduction.

We have also added the main advantages of a mechanical realization of a microwave circulator, in particular with respect to Josephson junction devices in the conclusion of the main text.

1.b It will be really helpful if the authors can estimate some parameters for experiment, i.e. temperature, cooperativity.

We thank the referee for raising this point. In the main text we added the measured refrigerator temperature of 10 mK, the measured thermalization temperature of the mechanical modes of (18 / 25) mK corresponding to a thermal phonon number of (85 / 90) for the mechanical modes 1 and 2. We also report the single cavity, single mechanical mode cooperativities for the pump powers used in the presented isolator and circulator experiments.

2a. What's the bandwidth of the isolator/circulator?

The instantaneous bandwidth of the device is a combination of the effective mechanical damping rates $\Gamma_{m,i}$ as defined in the equation C14 of the appendix. In our experiment they are given as $\Gamma_{m,1} = 2\pi 190$ Hz and $\Gamma_{m,2} = 2\pi 407$ Hz for the isolator and $\Gamma_{m,1} = 2\pi 209.9$ Hz and $\Gamma_{m,2} = 2\pi 624.9$ Hz for the circulator. The bandwidth of the system is given by $BW = \frac{4\Gamma_{m,1}\Gamma_{m,2}}{\Gamma_{m,1} + \Gamma_{m,2}}$ (see Ref [34] of the manuscript).

This results in a total bandwidth of ~ 518 Hz for the isolator and ~ 628 Hz for the circulator in agreement with the results presented in figures 2 and 3.

2b. Is there any more application of such small bandwidth especially for the microwave system?

Narrow bandwidth signal processing devices, such as the one presented, may still be used for example for narrow-band tunable filters and switches for effective noise rejection and sensitive precision measurements. On the other hand, we would like to point out that the shown bandwidth is not yet fundamentally limited at this point. It is a first proof of principle device. In fact, spurious mechanical modes prevented us to broaden the mechanics more using higher pump powers. Further design changes should improve the bandwidth to a level where it becomes more useful for circuit QED applications.

3. What's the input power to drive the mechanics and get higher cooperativity? Is there any nonlinear effect?

We have added the applied powers at the device inputs in the main text together with the cooperativities. In principle we would be able to apply up to about 10 dB higher powers before we observe nonlinear effects. This would greatly improve the bandwidth of the wavelength conversion and the isolator (using the 3rd cavity as a dissipative bath). However, the device showed additional weakly coupled mechanical

modes in the vicinity of the main modes whose origin is not entirely clear (potentially mixing with out of plane membrane modes). These additional modes make the interpretation of the measurement data very difficult so we chose to work at lower power.

4. *The figures are small and illegible to read. It can be improved to one row. And what's the unit of the Fig. 2b & 3b?*

We have increase the figure sizes and added the missing labels as recommended.

Reviewer #2 (Remarks to the Author):

In the manuscript "Mechanical On-Chip Microwave Circulator" the authors report on an experiment based on the existing theoretical ideas, to implement a microwave isolator/circulator. Integrating optomechanics with superconducting qubits, by using the same microwave technology, is interesting and could provide new possibilities. This work builds on the existing ideas and experiments in cavity optomechanics (see ref. 26 and refs within). The authors demonstrate that not only the operating frequency of the device can be tuned, but also the direction of the isolation/circulation can be controlled. The experimental results are in a good agreement with authors' theoretical analysis. I think if the authors address the issues below, publishing the paper in Nature communication can benefit a wide range of audience.

We thank referee for these positive comments.

1) *In the second paragraph, where limitations of the existing circulators based on magneto-optic effects have been discussed, no reference is presented.*

We have added two references [13][14] outlining the functionality and magnetic field requirements of ferrite based commercial circulators and revised the text to highlight current limitations of commercial circulators.

2) *I think it helps if the authors clarify what the optomechanical and electromechanical couplings refer to in their setup.*

We thank the referee for pointing this out. We have been using the terms optomechanical and electromechanical coupling interchangeably. In order to avoid any confusion we now only use the term electromechanical coupling in the manuscript. The electromechanical coupling between a single microwave resonator i and mechanical oscillator j is defined in the paragraph containing equation 1 of the main text as $G_{ij} = \sqrt{n_{ij}} g_{0,ij}$ with n_{ij} the number of drive photons in the system. Section 1 of the SI outlines the definition and calibrated values of the vacuum field electromechanical coupling $g_{0,ij}$.

3) *While the physics of non-reciprocity is based on reservoir engineering and the presence of mechanical loss, the intuitive picture behind non-reciprocal mechanism is postponed after the presentation of formalism and the results. I think it's helpful to present the physical picture first, and then delve into a mathematical description.*

As suggested we have added an intuitive explanation for the origin of nonreciprocity right before going into the theoretical details. We also revised the main text and now present the intuitive picture about the bandwidth of the isolator before presenting the results in the second paragraph of page 4.

4) *It seems that in insets of Fig 3b the black arrow indicates the circulation direction. It would be useful to mention it in the caption.*

We have implemented the suggested changes.

Reviewer #3 (Remarks to the Author):

The primary innovation in the paper "Mechanical On-Chip Microwave Circulator" is the first experimental electro-mechanical circulator that is potentially chip-compatible with other cryogenic microwave components. This is in contrast to the various demonstrations of opto- (refs 24-26) and electro-mechanical (refs 28 & 29, apparently 29 is now in print) isolators, which are two port non-reciprocal devices, as opposed to three.

Not without good reason, non-reciprocal components are having a bit of a "moment" right now in integrated micro- and quantum-devices in the optical and RF domains, due to the limitations of the standard technologies based on permanent magnets, as mentioned in the manuscript. Thus, this paper opens the door to a new technological approach (i.e. mechanical) to achieving non-reciprocity, and has the potential to be of interest to a fairly wide audience in integrated micro- and quantum-devices in the optical and RF domains. So I feel the topic matter and this paper in general deserves to be published in a broad journal. The main limitation of the paper is that the circulator's performance is extremely poor by almost every metric compared to already proven Joseph junction-based RF circulators (refs. 4 & 5 are two, but not the only examples), and it's not at all clear how the mechanical approach could ever out compete them. So, it deserves publication for its basic science, even if I am very pessimistic about its technological impact.

Assuming the authors can adequately address my questions and comments further below, I would generally support publication in Nature Communications.

We thank the referee for his positive comments and hope to be able to address all remaining questions in detail below.

1) *A small thing, but I generally prefer to describe devices such as this one as "electro-mechanics" not "opto-mechanics." Here the authors sort of use the terms interchangeably, which is not uncommon in the field because they are so closely linked, but I wonder if it might prove confusing to other readers.*

As suggested, we have removed all instances where we used the term optomechanics and replaced it with electromechanics.

2) *Fig. 1c needs to be redrawn to make the electrical connectivity much clearer. I cannot tell at all what is electrically connected to what in the nanowire device and the text didn't help me understand what I*

couldn't see in the figure. In particular, I cannot tell at all where the V_{dc} is applying its potential. I am very confused why "As expected, resonators 1 and 3 are tuned to higher frequency due to an increased vacuum gap while resonator 2 is tuned to lower frequency." As far as I can tell in Fig. 1c, resonators 2 & 3 look identical.

We thank the referee for this comment. As suggested, we have enlarged figure 1 and modified panel c (increased the capacitor gap sizes) to make the circuit connections clearer.

For this experiment, we use 4 capacitors on 2 nanostrings which by design move together as one mechanical mode of the nanobeam (the two nanostrings are part of one silicon beam). Applying a dc voltage on the left side of the upper nanostring displaces the entire nanobeam and reduces the capacitor gaps of both, the dc voltage capacitor on the left and the resonator 2 capacitor on the right. Conversely, the lower two capacitor gaps of resonators 1 and 3 are increased by the displacement of both nanostrings. This capacitance changes result in the observed shifts of the resonator frequencies.

3) In figure 1a, it looks like there are two additional inductively coupled ports. What are these for?

The two high frequency ports labeled 1 and 2 are formed by shorting a coplanar waveguide center conductor to the ground plane using thin wires appropriate for inductive coupling. Port 1 uses one such wire and port 2 uses two wires. This design was chosen in order to carefully control the extrinsic couplings and cross couplings to each of the 3 resonator modes used in the experiment. The only other connection, which is shown in the top left of panel 1a, is formed by two long wires that are connected to the two sides of the dc tuning capacitor. Here we apply a static voltage for tuning the resonator frequencies.

We have modified the caption to clarify the role of the different device connections.

4) Please state the overall footprint area of the device.

The total device area is 0.3 mm x 0.45 mm and we have added this information in the caption.

5) If the cooperativity employed in the main text figures was stated, I missed it. Similarly, please also state the pump power required to achieve this cooperativity in dBm. This required power metric is helpful in comparing this technology to others, such as the JJ devices.

We thank referee for raising this point. We have added the cooperativities and pump powers as suggested. Please also refer to our response to reviewer 1.

6) What is the relationship between the theory discussed at length in this paper to that described in the theory papers referenced in refs 6, 22, & 23?

In the mentioned earlier theory proposals the origin of nonreciprocity relies on a coherent coupling between the 2 resonator modes. In contrast, our experiment and theory makes use of 2 mechanical modes to realize an effective direct interaction and realize nonreciprocity. Furthermore we include all cross-coupling terms between all the involved mechanical and resonator modes. These off resonant couplings turn out to be essential to understand the bandwidth and the required detuning of the drive tones from the bare mechanical frequency. Finally, our theory (and experiment) is the first to include circulator physics with 3 resonator and 2 mechanical modes.

What is the relationship between the theory in this paper and the theory in the JJ-based components? For example, ref. 4 also involves parametric frequency conversion between the three port modes and two internal modes to achieve circulation. Is this approach and that one formally equivalent?

The underlying physics and theory of electromechanical systems is different from Josephson junction physics with different Hamiltonians, different coupling terms and different loss mechanisms. The parameter regime and approximations are different. For example the mechanical mode has finite occupation limiting the added noise but it is cooled when the device is turned on. In our case in particular, we use drive tones to couple two mechanical modes which in turn introduce an effective interaction between two of the cavities – an approach that to our knowledge has not been proposed or realized in JJ based systems.

Having said that, both the mechanical and the so far presented JJ cavity based approaches fundamentally require a dissipative bath to annihilate one of the propagation directions. Both approaches rely on interfering signal paths with carefully controlled phases as imposed by external pump tones. Also, the final form of the desired scattering matrix is of course the same for any circulator, but as we pointed out above, the physics to assemble this matrix is different.

7) I want to see not just the forward and backward scattering matrix elements, but S_{11} , S_{22} , & S_{33} as well. These input reflections are as important as the isolation for a usable device.

Figure 1 in the SI shows the performance of our device as a wavelength converter (reciprocal mode), where we obtain excellent agreement with theory for both transmission and reflection at both ports. In case of the isolator and circulator and having a detailed and matching theory, we distinguish between the internal resonator loss and insertion loss due to imperfect input matching. The insertion loss is between 3.8 dB and 4.4 dB for the 3 different ports as indicated in figures 2 and 3 panel b.

We agree that a full specification of a useful circulator ideally also includes detailed information about the reflection coefficients (in addition to the already specified impedance matching). However, the scattering parameters we measure and fit to theory clearly proves the directionality aspect of the circulator. We feel this is appropriate for a proof of principle circulator based on a new physical effect.

We have clarified the role and origin of the insertion loss due to finite input matching in the main text and increased the figure and label sizes indicating the different types of losses for better visibility.

8) Similarly, I want to see an output PSD of all three ports while the component is circulating. Again, as a practical matter, S-parameters are not the only thing you care about in a circulator, especially an active one. This device employs six pump tones that are orders of magnitude stronger than the signal tones and merely MHz from the signal carriers. It is critically important to know how bad all this RF leakage is.

Figure 3 in the SI depicts the noise properties of the system when there is no signal circulating (measured with the spectrum analyzer with all 6 pumps on but no signal tone applied). We believe that this shows that there is no RF leakage from the pumps present over the relevant band (other than the added noise shown). In the presence of a weak signal tone (weak compared to the power handling capabilities) the only difference would be a narrow peak in the spectrum (depending on the chosen signal frequency and power, and the chosen resolution bandwidth of the spectrum analyzer). This peak's magnitude is exactly what we measure when we show the S parameters in the manuscript (using a spectrum analyzer in zero span mode).

In the inset of figure 1 in the SI we show the signal power handling capabilities of the device used as a wavelength converter. For the shown S parameter measurements, we used signal powers of about -117 dBm well below the compression point. At these small signal powers, the phase noise is negligible compared to the noise induced by the (up to 6) pump tones whose relevant noise properties are shown in figure 3 in the SI.

We have added the typically used signal powers for the different experiments in the main text. We also clarified the meaning of figure 3 of the SI, i.e. that it includes all unwanted noises and potential spurious modes or RF leakage to be expected when the device is on (pumps are on) in the relevant section of the SI.

We agree that characterizing the noise properties of the device for higher signal powers would be interesting, but we feel like it goes beyond the scope of the current manuscript and would not affect its main findings.

9) I think it would benefit readers if the authors provided more of an honest assessment of this technology vis a vis other non-magnetic approaches. In particular, the JJ devices seems to out perform this one in just about every way, and it's not at all clear if this could ever change. For example, this device provided ~10 dB relative isolation over merely ~300 Hz, while ref. 4 (the first ever demonstration) had the same over 11 MHz. This component may be tuned by 30 MHz, while that one could be tuned up to 400 MHz. This one had ~4 quanta of added noise, while that one had 0.5 (quantum limited). This one requires six RF pumps, that one 3. Presumably the pump power required here is again orders of magnitude greater than in ref. 4, because the nonlinearity is so much weaker. If I had to guess, this mechanical technology might only be able to outperform the JJ ones in 1 dB compression point, again because the nonlinearity is so much weaker.

The referee raises some important points by comparing JJ devices to our mechanical version of a chip scale circulator. He/she is right that it will be difficult for mechanical systems to outperform JJ devices in terms of bandwidth. But in our opinions, it is too early to draw a definitive conclusion because in the end the bandwidth mainly depends on how strongly the system can be coupled to the dissipative bath.

In terms of added noise the mechanical systems need to be cooled to thermal occupations close to the ground state. The final occupancy for the reported device already reached as low as 0.6 quanta (for a single and strong cooling tone – the second mechanical mode reached an occupancy as low as 2 quanta). We are convinced that improved fabrication and design will lower the total added noise in the future to reach similar levels as JJ devices.

While our circulator required 6 pump tones because we work with 2 mechanical modes, improved designs with predictable direct resonator coupling will only require 3 pump tones.

While it might be challenging to catch up with existing technology on all fronts, we also see a number of unique advantages of mechanical systems compared to JJ technology:

- Mechanics is insensitive to magnetic field noise and offsets.
- As pointed out by the referee, the dynamic range can be higher (useful for example for high power dispersive qubit readout without a parametric amplifier). For the reported device operated as a wavelength converter no power saturation occurs for up to 10^4 cavity photons or about -90 dBm of signal power.

- Mechanical elements are well-localized objects where parasitic coupling across circuits on the same chip is reduced compared to electromagnetic modes.
- Devices based on mechanical oscillators have the potential for hybrid microwave and optical signal processing, in particular non-reciprocity between microwave and optical propagating fields.

We have added a concluding paragraph outlining the challenges and potential of our results in particular in comparison with Josephson junction devices.

As for the proposed application of superconducting qubit readout, the bandwidth of those signals are measured in MHz, and carriers can easily vary by 100 MHz without a very well controlled process. Even with its better parameters, the limitations of ref. 4 have yielded little actual technological impact so far for this application either. I want to reiterate that publishable work just has to be innovative, rather than technologically competitive, but the authors can only help the field by comparing it to other technologies and offering more guidance on where improvements will be best made.

We agree that the path to actual impactful applications can be a long one. Our main goal was to show important new physics and demonstrate what state of the art electromechanics can do today. This has been achieved in a new SOI platform that is compatible with both qubits and silicon photonics. In the next years we will work on making our circuits smaller, more integrated, more reliable and more useful.

REVIEWERS' COMMENTS:

Reviewer #1 (Remarks to the Author):

I am very happy with the revised manuscript as the authors have carefully addressed all the comments and suggestions raised in the reports. I have no further questions on it and therefore, I recommend it for consideration of publication in Nature Communications. No further review is necessary.

Reviewer #2 (Remarks to the Author):

My concerns are addressed. I recommend for publication.

Reviewer #3 (Remarks to the Author):

This is the second time I have reviewed the manuscript "Mechanical on-chip microwave circulator." I thank the authors for their in-depth reply and responsiveness to my and the other referee comments.

I think the paper is suitable for publication as long as they remove the last statement "Direct integration with superconducting qubits should allow for on-chip single photon routing..."

This component is orders upon orders upon orders of magnitude away from being suitable for this application. The -70 dBm CW pump tones mere MHz from the signal frequency will strongly drive any circuit QED readout cavity and completely blow away the qubit. Moreover, the 600 Hz bandwidth is more than 10^3 too restrictive for any cQED-appropriate signal routing. In contrast, the JJ-based circulators have less of a problem with this application because the pumps are much, much weaker, are GHz detuned, and have $\sim 10^8$ MHz of bandwidth. Readers (or the authors) should not be encouraged to integrate this device with a superconducting qubit.